# Exploring the Transcriptome Dynamics of In Vivo *Theileria annulata* Infection in Crossbred Cattle

**DOI:** 10.3390/genes14091663

**Published:** 2023-08-22

**Authors:** Sonika Ahlawat, Vikas Choudhary, Reena Arora, Ashish Kumar, Mandeep Kaur, Pooja Chhabra

**Affiliations:** 1ICAR-National Bureau of Animal Genetic Resources, Karnal 132001, Haryana, India; 2Department of Animal Husbandry and Dairying, Karnal 132001, Haryana, India

**Keywords:** differential expression, mRNA, parasite, RNAseq, theileriosis

## Abstract

The molecular changes occurring in the host in response to in vivo *Theileria annulata* parasitic infection are not well understood. Therefore, the present study investigated the differential expression profiles of peripheral blood mononuclear cells (PBMCs) across *Theileria annulata*-infected and non-infected crossbred cows. The differential expression profiles from PBMCs of infected and non-infected crossbred cows were generated by RNA sequencing. A marked difference in the expression of genes associated with innate immunity (*FTH1*, *ACTB*, *ISG15*) was observed between the two groups. The over-represented pathways in *Theileria annulata*-infected cows were associated with the immune system and regulation of the mitotic cycle. Enriched genes and pathways in non-infected animals were associated with the maintenance of chromatin integrity and cell structure. The highly connected genes identified in this study form potential candidates for further investigation into host–parasite interactions in cattle. An improved understanding of the transcriptomic dynamics during theileriosis would lead to underpinning molecular level differences related to the health status of cattle.

## 1. Introduction

India is the largest milk producer in the world, with the cattle population contributing to almost half the total produce [1]. Most of the milk comes from crossbred cattle as the indigenous breeds are low yielders. Theileriosis is a tick borne haemoprotozoan disease that affects 30–60% of crossbred (*Bos indicus* × *Bos taurus*) cattle in India [2]. *Theileria annulata* is transmitted through saliva by the ticks of Hylomma genus, present in the cattle feed [2]. Theileria belongs to the class of Apicomplexa parasites that have evolved sophisticated mechanisms to manipulate and evade the host cell machinery and immune response. The protozoan parasite invades the lymphocytes of the host, which results in progressive damage to the lymphoid system [3]. Major symptoms of theileriosis include an increase in body temperature, dullness, anorexia and decreased milk production. Acute theileriosis may result in severe loss of productivity and high mortality [2]. The problem is further confounded by resistance to drugs by the parasite. Since crossbred cattle contribute to 26% of the total milk produce in India [1], the economic impact of theileriosis is substantial. It has been reported that indigenous, *Bos indicus* cattle are more resistant to theileriosis than crossbred cattle [4,5].

The molecular changes occurring in the host in response to the parasitic infection are less understood. The next generation sequencing techniques allow the wholesome view of the gene expression in the host in comparison to microarray-based studies. The RNA sequencing technology has been widely used to illustrate the differential expression profile of genes in several parasitic diseases [4]. The use of comparative transcriptomics has opened the possibility of identifying regulatory elements underpinning the host response to infection. Recent progress has been made to understand the gene expression changes in response to Theileria infection in buffalo, native and crossbred cattle [4,5]. The majority of these studies have investigated the host–parasite interactions in vitro [4,6]. However, a comprehensive insight into the transcriptional dynamics of theileriosis occurring under field conditions is completely lacking. A comparative analysis of both groups will help in delineating important genes and pathways affected by the infection. The present study, therefore, investigated the differential expression profiles across *Theileria annulata*-infected and non-infected crossbred cows using RNA sequencing.

## 2. Materials and Methods

### 2.1. Samples

The study was duly approved by the Institutional Animal Ethics Committee, National Bureau of Animal Genetic Resources, Karnal. Blood samples were collected from female crossbred cattle (Holstein Friesian × Sahiwal; aged 4.5–5 years) from a single village in Karnal, Haryana, India. Infected animals exhibited symptoms such as pyrexia, swollen lymph node, anorexia and reduced milk production, while the non-infected animals showed no such symptoms. No experiments were performed on the animals during the study. The samples were collected by a trained veterinarian following all ethical norms during the handling of the animals. Three aliquots of each sample from infected and infection-free animals were made for microscopic examination, DNA isolation and RNA extraction, respectively. The samples were tested in the District Disease Diagnostic Laboratory, Karnal, Haryana, India.

### 2.2. Microscopic Examination

For blood microscopic examination, thin smears were prepared and stained with Giemsa stain in accordance with the standard protocol for haemoprotozoan identification [7]. A total of 75 microscopic fields were examined with a magnification of 100× for detection of the pathogen.

### 2.3. Molecular Detection of Haemoparasite in Blood

Genomic DNA was isolated from 200 μL of blood of each animal using the QIAamp DNA Kit (QIAGEN) as per the manufacturer’s protocol. The quality and concentration of the extracted DNA were ascertained using agarose gel electrophoresis (0.8%) and a Nanodrop spectrophotometer (ND-1000, Thermo Scientific). The pentaplex PCR assay was used to confirm the infection with *Theileria annulata* and to exclude infection from *Babesia bigemina*, *Trypanosoma evansi* and *Anaplasma marginale* in the samples [8].

### 2.4. RNA Isolation and Sequencing

Four samples that were found positive for *Theileria annulata* infection and four that were negative, based on the microscopic examination and PCR test, were further processed for RNA isolation (Figure 1). Peripheral blood mononuclear cells (PBMCs) were isolated from whole blood by density gradient centrifugation (Histopaque-1.077, Sigma). Briefly, 2.5 mL of peripheral blood was layered on 2.5 mL Histopaque and centrifuged at 500 g for 30 min. The white band of PBMCs was collected and washed twice with isotonic phosphate buffered saline (1× PBS). The cell pellet was dispensed in 1 mL of TRI Reagent (Sigma). Subsequently, total RNA was extracted using a standardized protocol using chloroform and isopropanol, followed by DNAase treatment. A bioanalyzer (Agilent Technologies) was used for determining the quality and concentration of the isolated RNA, and samples with a RIN value ≥7.0 were processed for library preparation. The libraries were prepared using NEBNext Ultra II RNA Library Prep Kit (Illumina) and 150 paired end RNA sequencing was performed on the Illumina HiSeq-2500 Platform.

### 2.5. Data Analysis

The quality of the raw RNAseq reads was assessed using FastQC-v 0.11.5 [9]. Data were analyzed using CLC Genomics Workbench 6.5.1 (CLC Bio, Aarhus, Denmark). The filtered reads were aligned to *Bos taurus* (ARS-UCD1.2) reference assembly. Gene expressions were normalized as reads per kilobase million (RPKM). Reads with RPKM values <0.01 were excluded from the study. The distribution of transcripts across two groups was visualized by a Venn diagram constructed using Venny [10]. The differential expression of genes was analyzed using the CLC transcriptomics analysis tool. Only those differentially expressed (DE) genes having log 2 fold change ≥2.0, an adjusted *p* value (p_adj_) <0.05 and an adjusted FDR <0.05 were subjected to further analyses. DAVID [11,12], Reactome [13] and Consensus Pathway Data Base [14,15] were used for the functional annotation and pathway analysis of the differentially expressed genes. A gene–protein network was constructed using Cytoscape ver 3.6.0 [16] and Cytohubba [17]. 

### 2.6. Validation by Real-Time Quantitative PCR (RT qPCR)

A real-time quantitative PCR was performed for validation of expression of the selected genes. Twelve genes, namely *ATN1*, *CAPG*, *CBX4*, *GPX7*, *ACTB*, *CDH1*, *CDK1*, *ISG15, MCM5, SRGN, TAP* and *TRIM25* and two reference genes (*RPLP0* and *RPS28*) were randomly selected for the RT qPCR. Primers sequences for these genes were synthesized using Primer 3 software [18] or based on previous publications (Appendix A). The RT qPCR reaction mix consisted of 2 µL of cDNA, 8 µL of qPCR master mix (5 µL of SYBR Green Real-Time master mix, 0.3 µL of each primer, 2.4 µL of DNA/RNA-free water) and was performed in triplicate on a Roche Light cycler 480 system. PCR efficiency was estimated by standard curve calculation using four points of cDNA serial dilutions. The mean cycle threshold (Ct) values of the genes were normalized to the geometric mean of the reference genes *RPS28* and *RPL0.* The data were analyzed by the 2^−ΔΔCT^ method [19].

## 3. Results

### 3.1. Summary of RNA Seq Data

The raw reads obtained for each sample ranged from 67 to 89 million (Table 1). The raw sequence data have been deposited in the NCBI Bioproject PRJNA702905 under accessions SAMN18116817–820 (infected) and SAMN17982889–892 (non-infected). Gene expression levels were normalized by counting the number of RPKM (reads per kilobase of transcript per million mapped reads). The processed reads were mapped to the *Bos taurus* (ARS-UCD1.2) reference assembly with a mapping percentage of >94 for each sample. A cutoff value of 0.01 RPKM was used for further analysis. The total genes identified in the uninfected and infected samples were 15,200 and 14,413, respectively (Appendix A). Unique transcripts discovered in the control animals were 1147, and 360 were unique to infected animals while 14053 transcripts were common to both groups (Figure 2).

### 3.2. Gene Expression Profile

The top 25 highest expressed genes with RPKM > 2000 identified in both groups are shown in Figure 3. The genes, namely, *RPLP1*, *COX1*, *TPT1* and *COX3* exhibited maximum expression at >5000 RPKM. A gene ontology analysis delineated these genes into major functional categories such as the ribosomal proteins, ribonucleoprotein, phosphoprotein and acetylation. The enriched biological processes were translation, ribosomal small subunit assembly, antigen processing and presentation of peptide antigen via MHC class I, while the significant cellular components were extracellular exosome, cytosolic small ribosomal subunit and focal adhesion (Figure 4, Appendix A). The molecular factors represented by the highly expressed genes included the structural constituent of ribosome, poly (A) RNA binding and cytochrome-c oxidase activity. Significant differences were observed in the expression of *FTH1*, *ACTB*, *ISG15* and *S100A12* genes in the control and infected cows. These genes were up-regulated in animals infected with Theileria in comparison to non-infected animals.

### 3.3. Validation of RNAseq Data by RT-qPCR

The genes for validation by RT-qPCR were randomly chosen from the significantly differentially expressed genes (p_adj_ < 0.05) both up-regulated and down-regulated in infected animals. The relative levels of expression of the selected genes (*ATN1*, *CAPG*, *CBX4, GPX7, ACTB, CDH1, CDK1, ISG15, MCM5, SRGN, TAP* and *TRIM25*) were concordant with the expression levels detected by RNAseq data (Appendix A), thus validating the RNA sequencing information.

### 3.4. Differentially Expressed Genes and Enriched Pathways

A comparison of the gene expression between the infected and non-infected individuals led to the identification of 586 differentially expressed genes with p_(adj)_ and FDR <0.05 and a log_10_ fold change of >±2.0. Of these 436, were up-regulated, while 150 were down-regulated in the infected animals (Appendix A). Assessing the functional relevance revealed that the major biological processes represented by these genes included immune response, defense response to virus, bacterium, chemotaxis and positive regulation of cell proliferation (Figure 5, Appendix A). The enriched cellular components were extracellular region and space, while the molecular factors included serine-type endopeptidase. The 25 most relevant pathways in the infected and non-infected group, with respect to p values, are given in Appendix A. The significant pathways identified for the differentially expressed up-regulated genes in the Theileria-infected group included interleukin-10 signaling, immune signaling, cytokine signaling in immune system, interferon signaling, cell cycle RHO GTPases activate formins and the ISG15 antiviral mechanism. Amongst the pathways representing the uninfected group, RNA polymerase 1 promoter opening, DNA methylation, HDACs deacetylate histones, senescence-associated secretory phenotype (SASP) and DNA damage/telomere stress induced senescence were significant.

### 3.5. Gene–Protein Interactions

In order to identify the highly connected or hub genes, a gene–protein interaction network was constructed from the differentially expressed genes. The network was constructed by selecting differentially expressed genes having equal to or more than 5 interactions. The top 10 genes with the maximum interactions were identified as important hub genes from the network. The network for up-regulated genes in infected cows comprised 89 nodes and 234 edges, while that in non-infected animals had 67 nodes and 119 edges. The highly connected up-regulated genes in the infected group were *TRIM25, CDK1*, *TAP*, *AURKB*, *KIF20A*, *MKI67*, *CDH1*, *CDC20*, *AURKA* and *NDC80* (Figure 6). Among the differentially expressed up-regulated genes in the non-infected animals, *H2AC4*, *SNCA*, *H3C13*, *MCM5*, *H1–4*, *STX1A*, *EFEMP1*, *ATN1*, *DSP* and *HIST2H2AA3* ranked as the top 10 most important genes (Figure 7).

## 4. Discussion

The comparative analysis of infected and non-infected animals in our study revealed that the majority of the common genes with high expression in both groups included ribosomal protein and cytochrome c oxidase genes (*RPLP1*, *RPL13A*, *RPS15*, *RPS11*, *RPS3A, RPS3*, *RPS20*, *RPL23*, *RPL37*, *RPS10*, *RPS5*, *RPS15A*, *RPLP0*, *RPS25*, *COX1*, *COX2*, *COX3*). These genes are essentially required for ribosome biosynthesis leading to protein synthesis and oxidative phosphorylation for energy production in all cells. The equally high expression of these genes in both groups reinforces their basic biological or housekeeping role in cell maintenance.

However, marked differences in the expression of *FTH1*, *ACTB*, *ISG15*, *S100A12* and *SRGN* genes were observed between the infected and non-infected cows. It is well-established that both innate and adaptive immunity contribute towards defending the host against parasitic infections [20]. The transcriptomic comparison of infected and healthy animals in our study revealed that genes associated with immune response were over-expressed in infected cows. Interferons (IFNs) form part of both the innate and adaptive immune response. The interferon family equips the host with a primary defense mechanism against the pathogens by up-regulating interferon stimulated genes (*ISG*s) [21]. The *ISG15* gene in particular is most effectively activated by IFNγ in humans infected with protozoan parasites [22]. *ISG15* has also been reported to be up-regulated in bovine Theileria-infected cells [23]. Serglycin encoded by the *SRGN* gene is a major proteoglycan known to be involved in immune response in mammals [24]. The expression of *SRGN* has been reported to be up-regulated in sheep infected with the *Haemonchus contortus* parasite [25]. Both *FTH1* and *ISG15* genes are activated in response to pro-inflammatory cytokines mediated by the host defense mechanism. The presence of IL-1β and IL6 activates the expression of *FTH1* through the NF-Κb pathway [26]. Studies have indicated that the up-regulation of *FTH1* is implicated in cell proliferation and apoptosis [27]. The parasite, associated with the spindle apparatus of the host cell, induces the proliferation and transformation of the host cell [20]. Although there is inadequate information on the function of *S100A12* in parasitic infection in livestock, it is known to be highly expressed in bacterial infection in humans [28,29]. *ACTB* on the other hand, is a housekeeping gene involved in cytokinesis and maintaining cell shape [30]. The higher expression of *FTH1, ISG15, SRGN* and *S100A12* genes observed in infected animals suggests the activation of the host immune system to resist the infection.

As expected, the over-represented pathways in Theileria-infected cows were mainly associated with the immune system and regulation of the mitotic cycle (mitotic metaphase and anaphase). It is interesting to note that among the non-infected animals, pathways related to histone deacetylases (HDACs), telomerase maintenance as well as nucleosome assembly were enriched. These enriched pathways suggest the maintenance of chromatin integrity, which may play a protective role in the host cell [31].

The gene–protein network analysis led to the identification of important up-regulated genes in the infected as well as non-infected cows. The top 10 highly connected genes in infected animals were involved in interferon signaling and the cell cycle. Among these, *TRIM25* and *TAP* are associated with interferons, while *CDK1, AURKA, AURKB, KIF20A, MKI67*, *CDH1*, *CDC20* and *NDC80* are involved in cell cycle and proliferation. Invasion by a parasite prompts the innate immune response of the host to express cytokines such as IFNs and TNFs. The transporter associated with antigen-processing (*TAP*) and *TRIM25* genes are induced by interferons as an innate immune response [22]. *TAP* has been reported to be up-regulated in response to the expression of IFNs [32]. The *TRIM25* gene belongs to the tripartite motif (TRIM) protein family. Although the role of *TRIM25* in parasitic infection has not been explored, its expression is reported to be enhanced during mycobacterium infection [33].

Cell division is an intricate process coordinated by several genes and regulatory molecules. Cell cycle progression is regulated by cyclin-dependent kinase (*CDK1*), cell-division cycle protein 20 (*CDC20*) and *CDH1* which is a homologue of *CDC20* [34]. Previous studies have demonstrated that *CDK1* forms the target for interaction in cells infected with *Theileria annulata* [35]. Aurora kinases (*AURKA* and *AURKB*) belong to the family of serine and threonine kinase, which are involved in the regulation of cytokinesis [36]. The Aurora kinases as well as *KIF20A/MKLP2* regulate the spindle formation and are associated with the condensation of chromosomes and cell division [36,37]. These have been observed to be conserved across different parasites including Theileria [38]. The *NDC80* gene is involved in microtubule binding during mitosis [39]. Recent studies have also highlighted the importance of *NDC80*, *CDK1* and *KIF20A* genes in cell division during human diseases [40,41]. The presence of the *MKI67* gene that codes for Ki-67 and is used as a marker for proliferation [42] further corroborates that the Theileria parasite transforms the cell cycle machinery of the host, resulting in uncontrolled cell proliferation which is characteristic of theileriosis [43]. 

A majority of the highly connected genes up-regulated in infection-free animals were histone coding or associated with histones. The *H2AC4*, *H3C13*, *HIST2H2AA3* and *H1–4* are involved in the neutrophil extracellular trap (NET) pathway [44]. NETs are derived from neutrophils as a defence mechanism to trap and destroy the pathogen. NETs are composed of DNA, proteinases and histones. Emerging evidence supports the role of histones in anti-parasitic activity as an innate immune defense mechanism [45,46]. Although direct evidence of the involvement of histone proteins in host defense against theileriosis is not known, they are reported to demonstrate antiparasitic activity [44]. The enrichment of such genes in the non-infected animals suggests a better protection mechanism against the parasite.

Sufficient information on the role of *MCM5*, *ATN1*, *SCNA* and *STX1A* genes in the context of parasitic infection is not available. However, mini-chromosome maintenance complex component 5 (*MCM5*) is involved in DNA replication and also plays a role in transcriptional activation in response to IFN-γ mediated by *Stat1α* [47]. The Atrophin1 (*ATN1*) gene is established as a transcriptional co-repressor that interacts with DNA binding proteins to suppress gene activity. Converging evidence suggests that Atrophin 1, in coordination with the Notch-mediated signaling pathway, regulates immune responses [48,49]. The α-synuclein protein coded by *SCNA* binds to fatty acids and is involved in pre-synaptic vesicle trafficking. Modifications in the α-synuclein protein have also been implicated in initiating an innate immune response [50]. The *STX1A* gene belongs to the syntaxin family and is involved in synaptic transmission and vesicle fusion [51]. Although the role of *STX1A* is still not clear, research evidence has implicated other members of the syntaxin family in phagocytosis [52]. 

Two of the hub genes in non-infected cows, *EFEMP1* and *DSP*, support cell structure. *EFEMP1*/Fibulin 3 is a suppressor of ECM proteolysis/degradation and maintains tissue integrity [53], and similarly, *DSP* provides mechanical strength to cells [54]. These genes over-expressed in non-infected animals probably help in maintaining the cell integrity and may impart protection to the cell against the parasitic infection.

## 5. Conclusions

Our results provide insight into the host gene expression in response to parasitic infection. The infected animals exhibited the activation of genes involved in immune response, while the healthy controls showed a higher expression of genes associated with histones. The enrichment of HDACs’ deacetylate histone pathway in non-infected animals suggests a state of increased condensation of the host DNA [55], thereby preventing access to the host DNA. Previous studies have demonstrated that the use of HDAC inhibitors suppresses the host’s defense mechanism [56]. These genes and pathways showed a lower or suppressed expression in the infected animals. The highly connected genes identified in this study form potential candidates for further investigation into host–parasite interactions in cattle.

## Figures and Tables

**Figure 1 genes-14-01663-f001:**
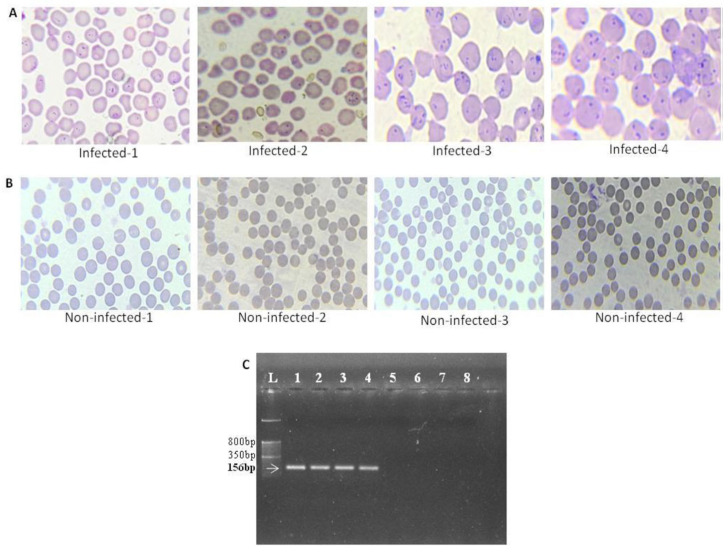
(**A**) Giemsa-stained blood smear showing presence of *Theileria annulata;* (**B**) Giemsa-stained blood smear showing absence of *T. annulata;* (**C**) Pentaplex PCR amplification, L-50 bp DNA ladder; lanes 1–4: 156 bp band confirming infection by *T. annulata*; lanes 5–8: no amplification confirming absence of *T. annulata*.

**Figure 2 genes-14-01663-f002:**
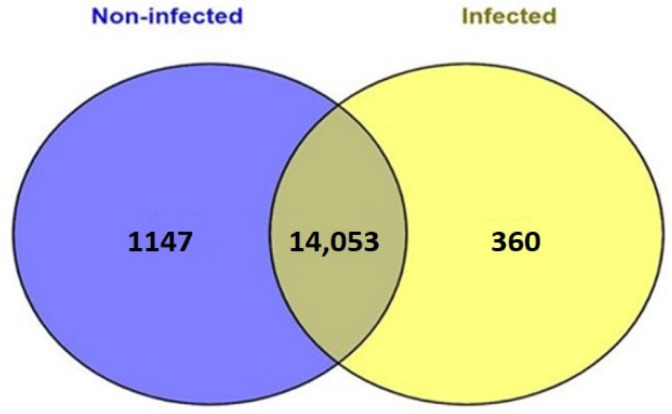
Venn diagram illustrating distribution of transcripts between infected and infection-free crossbred cows.

**Figure 3 genes-14-01663-f003:**
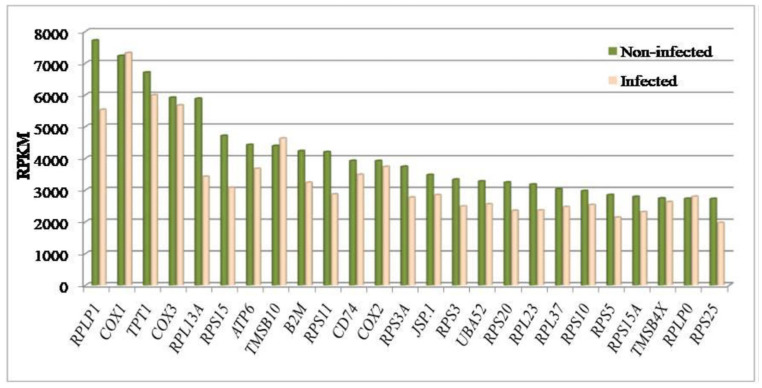
The top 25 highest expressed genes (RPKM > 2000) across both infected and non-infected crossbred cows.

**Figure 4 genes-14-01663-f004:**
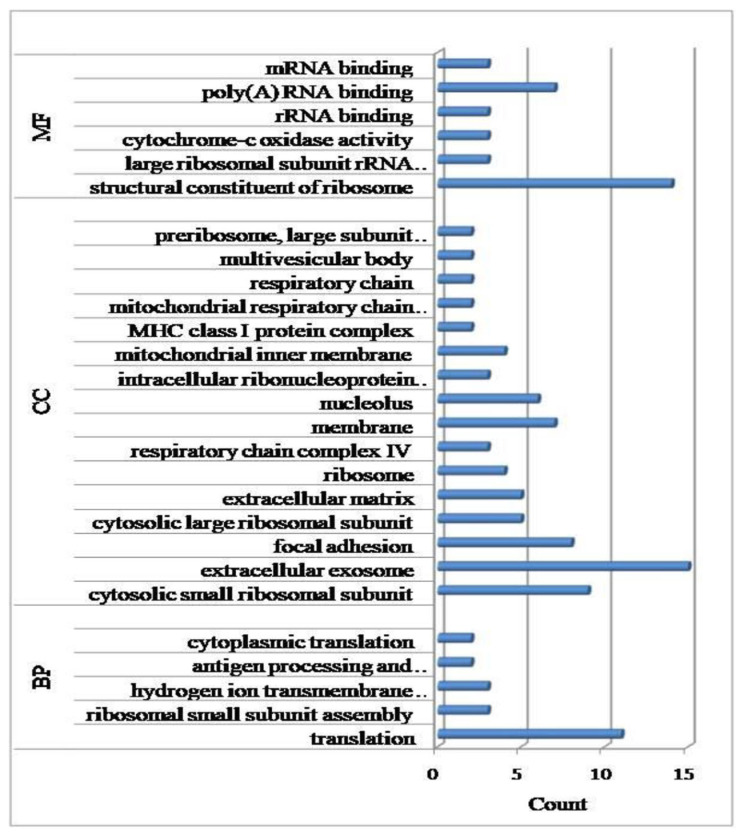
Gene ontology analysis of top 25 highest expressed genes common to infected and non-infected cattle (BP: biological process; CC: cellular component; MF: molecular function).

**Figure 5 genes-14-01663-f005:**
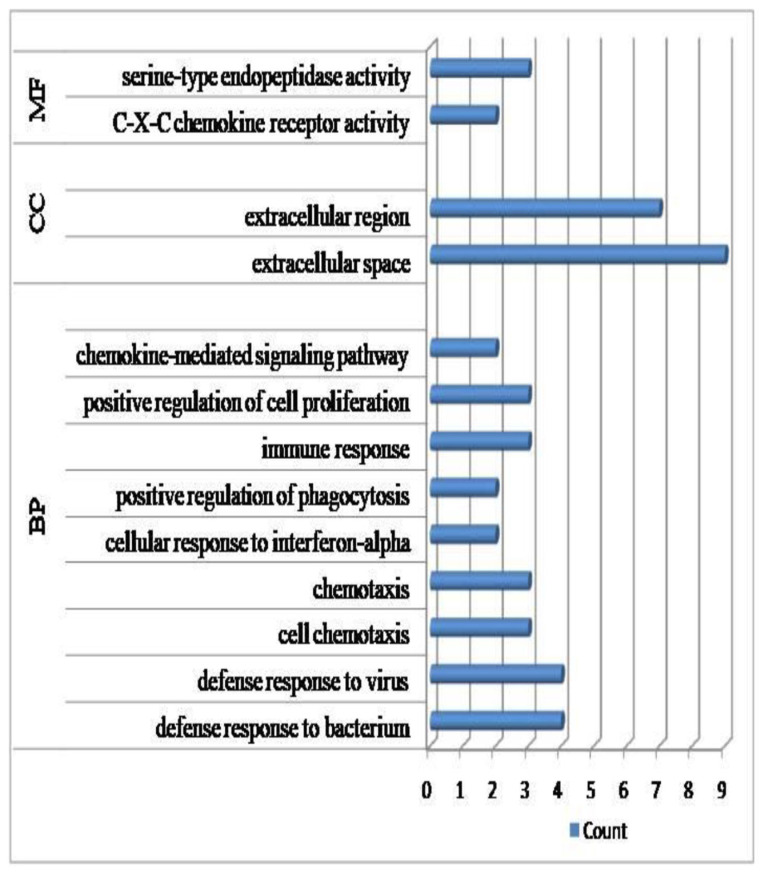
Gene ontology analysis of significantly differentially expressed genes across infected and non-infected cattle (BP: biological process; CC: cellular component; MF: molecular function).

**Figure 6 genes-14-01663-f006:**
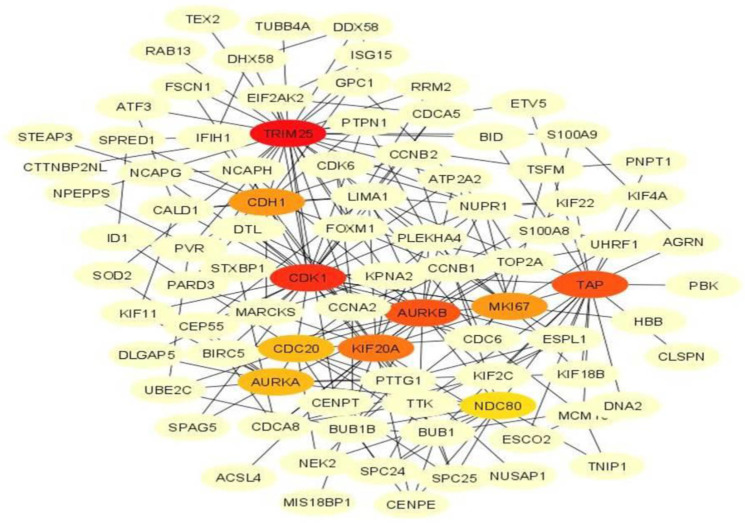
The highly connected up-regulated genes in *Theileria annulata*-infected animals. Color intensity of top 10 genes changes from orange to red with increasing order of rank.

**Figure 7 genes-14-01663-f007:**
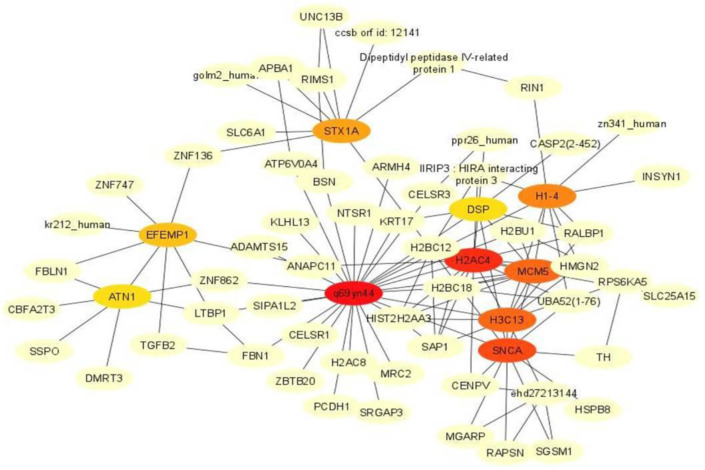
The highly connected up-regulated genes in non-infected animals based on degree of connectivity. Color intensity of top 10 genes changes from orange to red with increasing order of rank.

**Table 1 genes-14-01663-t001:** Reads’ summary and mapping rate for RNAseq data generated for *Theileria annulata*-infected and non-infected cows.

Crossbred Cattle Sample	Raw Reads	Processed Reads	Mapped Reads	Mapping %
Non-infected-1	67,187,256	66,822,938	65,686,885	98.30%
Non-infected-2	78,767,458	78,725,278	76,814,152	97.57%
Non-infected-3	86,213,224	86,172,538	84,479,731	98.04%
Non-infected-4	89,485,542	89,167,234	88,175,002	98.89%
Infected-1	84,605,308	84,572,394	80,296,629	94.94%
Infected-2	81,213,906	81,200,906	80,016,469	98.53%
Infected-3	79,494,742	79,400,742	78,529,749	98.79%
Infected-4	86,943,534	86,910,534	85,711,712	98.58%

## Data Availability

The data presented in this study are available on request from the corresponding author.

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
