# Peer review of "Exploring the Transcriptome Dynamics of In Vivo Theileria annulata Infection in Crossbred Cattle"

_genes, 2023, doi:10.3390/genes14091663_

Round 1
Reviewer 1 Report
The manuscript report important data regarding the analysis of the transcriptome from the monocytes obtained from peripheral blood of cattle infected with Theileria annullata and compared those significantly up/down regulated genes against non-infected cattle; this is an important study because of the cattle-parasite’s wide distribution and prevalence in many parts of the world.
The manuscript is well written and the experimental designs and execution is scientifically solid, however, the provided supplemental material only contains the names and statistical parameters of those genes that significantly expressed differences as well as the primers sequences for the qPCR corroboration, the authors did not place the transcriptome in a public database for free access to the obtained sequences, this is important because a deeper bioinformatic analysis may find that some of these genes show some SNTP and/or amino acid substitutions and/or alleles in the expressed up/down regulated peptides that may contribute to cattle resistance against theilerosis.
If the authors do not want to make public the transcriptome sequences, I suggest to upload to the Genbank the 12 individual sequences that exhibited statistically significant differential expression it will take only a couple of days and accession numbers will be provided immediately by the Genbank, although public accession will take some days, it is necessary for a follow-up independent bioinformatic analysis.
The Gene Ontology Enrichment analysis (GOEA) provided the Genbank/Uniprot accession numbers of each and every protein sequences with high identity score, this accession should be described with the genes names; ideally, this should include the accession numbers of the very same sequences identified by the transcriptome obtained during this study, however, they are not included in the manuscript and/or the supplemental material, I consider it should be in an additional table within the supplemental material.
The qPCR primers were designed using previously reported sequences that show genbank accession numbers the were not indicated in the M&M section; it is not clear why the author did not used their own sequences (with some exceptions) obtained during this study.
The discussion section is abundant in analysis and bibliographic description of the differentially expressed genes, however the conclusion are very succinct and I believe it should at least reflect the points highlighted in the previous section.
Author Response
Response to Reviewer 1 Comments
Point 1.The manuscript is well written and the experimental designs and execution is scientifically solid, however, the provided supplemental material only contains the names and statistical parameters of those genes that significantly expressed differences as well as the primers sequences for the qPCR corroboration, the authors did not place the transcriptome in a public database for free access to the obtained sequences, this is important because a deeper bioinformatic analysis may find that some of these genes show some SNTP and/or amino acid substitutions and/or alleles in the expressed up/down regulated peptides that may contribute to cattle resistance against theilerosis.
If the authors do not want to make public the transcriptome sequences, I suggest to upload to the Genbank the 12 individual sequences that exhibited statistically significant differential expression it will take only a couple of days and accession numbers will be provided immediately by the Genbank, although public accession will take some days, it is necessary for a follow-up independent bioinformatic analysis.
Response 1: The data is accessible under NCBI Bioproject PRJNA702905 under accessions SAMN18116817-820 (infected) and SAMN17982889-892 (non-infected).
Point 2. The Gene Ontology Enrichment analysis (GOEA) provided the Genbank/Uniprot accession numbers of each and every protein sequences with high identity score, this accession should be described with the genes names; ideally, this should include the accession numbers of the very same sequences identified by the transcriptome obtained during this study, however, they are not included in the manuscript and/or the supplemental material, I consider it should be in an additional table within the supplemental material.
Response 2: The details of the Gene Ontology analysis for Figures 4 and 5 have now been added as supplementary Table S4. The previous Table S4 is now Table S5.
Point 3. The qPCR primers were designed using previously reported sequences that show genbank accession numbers the were not indicated in the M&M section; it is not clear why the author did not used their own sequences (with some exceptions) obtained during this study.
Response 3: The qPCR primers used in previous studies were used as they were already tested, some of the primers were also designed in our study.
Point 4.The discussion section is abundant in analysis and bibliographic description of the differentially expressed genes, however the conclusion are very succinct and I believe it should at least reflect the points highlighted in the previous section.
Response 4: The conclusion has now been revised as suggested.
We thank the reviewer for valuable suggestions and trust that all the necessary changes have been incorporated.
Reena Arora
Reviewer 2 Report
Abstract
"The present study investigated the differential expression profiles across Theileria annulata infected and non-infected crossbred cows.".... expression profile of whom???? please add details.
This section is too long. Make it simple and too the point as standard abstract... what was aim? what were methods? what were major finidngs? what is your conclusion? Please do not discuss or justify the results in abstract.
The study is interesting as limited data is available on this topic. However, i have the following queries and suggestions for the authors to improve the manuscript further.
How the genes were selected for expression analysis?
Why only crossbred cattle were included in studies while India has local as well as exotic cattle breeds?
Again, while Theileria annulata mediated infection was targeted? A number of tick/vector borne cattle diseases are reported from India? What is the prevalence of this parasite in in Indian cattle or specifically in crossbred cattles?
How the Theileria annulata infection was confirmed in enrolled cattle? Please explain the pentaplex PCR details and share primer and annealing conditions as supplementary material, if not in main text.
Fig. 1A. what was the magnification of micro photographs included in the study?
Only 4 parasite positive and 4 negative samples are used for comparison. How will you justify that this number is enough to make reliable conclusions?
Fig. 3. Was the gene expression compared statistically between infected and non infected cattle? If yes, please add it in caption or in foot note.
There are no results of penta plex PCR other than Theileria annulata. Have the authors detected other parasites as well? If yes and if the cattle enrolled in studies we coinfected, how the authors can claim that the mentioned changes in gene expression are during Theileria annulata infection specifically?
If similar studies has been conducted in India or elsewgere previously then discuss them in comparision with the findings of current investigation.
Author Response
Response to Reviewer 2 Comments
Point 1. "The present study investigated the differential expression profiles across Theileria annulata infected and non-infected crossbred cows.".... expression profile of whom???? please add details.
Response 1: The Abstract has been modified as suggested.
Point 2. This section is too long. Make it simple and too the point as standard abstract... what was aim? what were methods? what were major finidngs? what is your conclusion? Please do not discuss or justify the results in abstract.
Response 2: Modified as suggested.
Point 3. How the genes were selected for expression analysis?
Response 3: The genes were randomly selected to cover both up - and down-regulation, as well as to include a wide range of expression levels.
Point 4 Why only crossbred cattle were included in studies while India has local as well as exotic cattle breeds?
Response 4: Cross-bred (Bos indicus x Bos taurus) cattle were selected for the study since theileriosis affects 30-60% of cross bred cattle in India. Bos indicus cattle, on the other hand have been reported to be more resistant to theileriosis than crossbred cattle.
Point 5. Again, while Theileria annulata mediated infection was targeted? A number of tick/vector borne cattle diseases are reported from India? What is the prevalence of this parasite in in Indian cattle or specifically in crossbred cattles?
Response 5: Theileriosis affects 30-60% of cross-bred (Bos indicus x Bos taurus) cattle in India (already mentioned in Introduction section).
Point 6. How the Theileria annulata infection was confirmed in enrolled cattle? Please explain the pentaplex PCR details and share primer and annealing conditions as supplementary material, if not in main text.
Response 6: The pentaplex PCR targets four parasites namely Babesia bigemina, Theileria annulata, Anaplasma marginale and Trypanosoma evansi in cattle. The details of the Pentaplex PCR are elaborated in the publication- Ganguly A, Maharana BR, Ganguly I. Pentaplex PCR assay for rapid differential detection of Babesia bigemina, Theileria annulata, Anaplasma marginale and Trypanosoma evansi in cattle. Biologicals. 2020 Jan;63:81-88. doi: 10.1016/j.biologicals.2019.10.011. This publication has been referred to in the manuscript, therefore, a detailed description has not been included..
Point 7. Fig. 1A. what was the magnification of micro photographs included in the study?
Response 7: The samples were examined under oil immersion lens (1000x). Objective lens is 10x and oil immersion 100x.
Point 8. Only 4 parasite positive and 4 negative samples are used for comparison. How will you justify that this number is enough to make reliable conclusions?
Response 8: As per ENCODE guidelines* two or more biological replicates can be used for RNA sequencing experiments. For a sample 15 million reads is considered sufficient, if number of replicates is more than 3. We used four biological replicates as per read depth and budget requirements with a longer read length (150 PE) and were able to generate more than 67 million reads per sample. About 30-60 million reads are considered adequate to provide differential gene expression and detection genes that have low expression.
*https://www.encodeproject.org/documents/cede0cbe-d324-4ce7-ace4-f0c3eddf5972/@@download/attachment/ENCODE%20Best%20Practices%20for%20RNA_v2.pdf
Point 9. Fig. 3. Was the gene expression compared statistically between infected and non infected cattle?
Response 9: Since the raw read counts are influenced by transcript length as well as other sequencing biases. Therefore to eliminate the errors due to library size or transcript length, the reads were normalized by the RPKM (reads per kilobase of exon model per million reads) method.* We have already mentioned in the Materials and Methods sections that ‘Gene expressions were normalized as Reads per Kilobase of transcript per Million mapped reads (RPKM).’ Besides, the expression levels of some genes have also been validated by quantitative real time PCR.
* Conesa A, Madrigal P, Tarazona S, et al. (2016) A survey of best practices for RNA-seq data analysis. Genome Biol.17:13. doi:10.1186/s13059-016-0881-8
Point 10. There are no results of penta plex PCR other than Theileria annulata. Have the authors detected other parasites as well? If yes and if the cattle enrolled in studies we coinfected, how the authors can claim that the mentioned changes in gene expression are during Theileria annulata infection specifically?
Response 10: The pentaplex PCR targets four parasites namely Babesia bigemina, Theileria annulata, Anaplasma marginale and Trypanosoma evansi in a single reaction. We observed a single band corresponding to T. annulata ruling out infection from the other three parasites.
Point 11. If similar studies has been conducted in India or elsewgere previously then discuss them in comparision with the findings of current investigation.
Response11: The studies previously available on Indian and exotic cattle are mainly on in vitro infection and have been mentioned in the Introduction. However, since our study is on live animals, a direct comparison was not feasible.
We thank the reviewer for valuable suggestions and trust that all the necessary changes have been incorporated.
Reena Arora
Round 2
Reviewer 2 Report
No further comments